# Exploring RNA-Seq Data Analysis Through Visualization Techniques and Tools: A Systematic Review of Opportunities and Limitations for Clinical Applications

**DOI:** 10.3390/bioengineering12010056

**Published:** 2025-01-12

**Authors:** Farhana Manzoor, Cyruss A. Tsurgeon, Vibhuti Gupta

**Affiliations:** 1Department of Computer Science and Data Science, School of Applied Computational Sciences, Meharry Medical College, Nashville, TN 37208, USA; fmanzoor24@mmc.edu; 2Department of Biomedical Data Science, School of Applied Computational Sciences, Meharry Medical College, Nashville, TN 37208, USA; ctsurgeon07@mmc.edu

**Keywords:** RNA-seq, sequencing, visualization

## Abstract

RNA sequencing (RNA-seq) has emerged as a prominent resource for transcriptomic analysis due to its ability to measure gene expression in a highly sensitive and accurate manner. With the increasing availability of RNA-seq data analysis from clinical studies and patient samples, the development of effective visualization tools for RNA-seq analysis has become increasingly important to help clinicians and biomedical researchers better understand the complex patterns of gene expression associated with health and disease. This review aims to outline the current state-of-the-art data visualization techniques and tools commonly used to frame clinical inferences from RNA-seq data and point out their benefits, applications, and limitations. A systematic review of English articles using PubMed, Scopus, Web of Science, and IEEE Xplore databases was performed. Search terms included “RNA-seq”, “visualization”, “plots”, and “clinical”. Only full-text studies reported between 2017 and 2024 were included for analysis. Following PRISMA guidelines, a total of 126 studies were identified, of which 33 studies met the inclusion criteria. We found that 18% of studies have visualization techniques and tools for circular RNA-seq data, 56% for single-cell RNA-seq data, 23% for bulk RNA-seq data, and 3% for long non-coding RNA-seq data. Overall, this review provides a comprehensive overview of the common visualization tools and their potential applications, which is a useful resource for researchers and clinicians interested in using RNA-seq data for various clinical purposes (e.g., diagnosis or prognosis).

## 1. Introduction

Every somatic cell in our body contains the same set of genes encoded in our genome, with the exception of gametes and certain specialized cells (i.e., red blood cells, gametes, and some immune cells etc.). However, not all genes are active or expressed at the same time, as gene expression varies depending on the cell type and its function. Each cell type, depending on its conditions or stage of development, expresses a unique subset of RNA transcripts. These include both protein-coding RNA (translated into protein) and non-coding RNA (with regulatory or structural roles instead of being translated into protein). The specific genes that are activated or deactivated define a cell’s function and offer insights into health and disease. Profiling the dynamic set of RNA transcripts, including both coding and non-coding RNA, has been a central focus of research for many years. Today, a common method for analyzing these diverse and distinguishing RNA molecules is RNA sequencing (RNA-seq).

### 1.1. RNA Sequencing

RNA-seq combines the study of natural biological processes (such as RNA transcription and degradation), experimental laboratory techniques (including RNA extraction, library preparation, and sequencing), and computational analyses (for transcript quantification, differential expression analysis, and functional annotation). It enables the sequencing and characterization of the complete set of RNA transcripts in a cell or tissue at a specific point in time, collectively known as the transcriptome [1]. At its core, RNA-seq is a method that utilizes massively parallel sequencing, commonly known as next-generation sequencing (NGS), to examine the repertoire of RNA molecules in a sample. NGS offers ultra-high throughput, scalability, and speed, making it an ideal platform for RNA-seq, which extends these capabilities to rapidly sequence and analyze millions of RNA transcripts. Through NGS, RNA-seq aims to capture a snapshot of dynamic gene expression by characterizing the present RNA and quantifying the abundance of active transcriptional processes at a given point in time or under specific physiologic conditions. RNA-seq has become a very popular tool for analyzing gene expression and enables the examination of diverse gene expression phenomena such as alternative splicing, post-transcriptional modifications, gene fusion events, mutations or SNPs, temporal gene expression changes, or variances of gene expression across different conditions or treatments. It is a transformative instrument for transcriptomics—the analysis of the transcriptome—and stands out as a revolutionary technique to decipher the molecular underpinnings of our genetic constitution.

RNA-seq has transformed the field of genomics since its inception by enabling the sequencing of entire transcriptomes at a fraction of the time and cost of earlier methods. Its growing accessibility and versatility has made it increasingly popular in genomics and molecular biology, facilitating the study of a broader range of RNAs and providing a deeper understanding of non-coding genomic sequences. RNA-seq is often selected over traditional methods for gene expression profiling, including reverse transcription-quantitative polymerase chain reaction (RT-qPCR), microarrays, serial analysis of gene expression (SAGE), expressed sequence tags (EST), and Northern blotting, due to its significant advantages for both research and clinical applications [2]. RNA-seq enables the identification and quantification of both known and novel transcripts [3]. This powerful technique revolutionized our understanding of gene regulation, signaling pathways, and cellular complexity by uncovering non-coding RNA functions, alternative splicing events, and post-transcriptional modifications. These advancements unraveled cellular processes on an unprecedented scale, making RNA-seq a cornerstone of modern transcriptomics [4].

RNA-seq has become an indispensable tool for analyzing gene expression, with various approaches tailored to specific research applications. Table 1 summarizes these RNA-seq approaches and their unique advantages. Bulk RNA-seq is the most common and well-established approach, measuring the average gene expression across a population of cells. It is ideal for broad-scale gene expression analysis and remains a standard for many studies [5]. Some studies will examine the entire transcriptome, while others will be more focused on only the coding RNA. Single-cell RNA-seq (scRNA-seq), by contrast, measures gene expression at the individual cell level, offering valuable insights into cellular heterogeneity. This approach is particularly important for studying complex tissues, such as the immune system or cancer microenvironments, where individual cell differences drive biological outcomes [6]. Spatial transcriptomics builds upon scRNA-seq with the addition of spatial context by mapping gene expression to specific locations within a tissue. This approach enables researchers to explore the spatial organization of cells and their interactions within the tissue [5]. Long-read RNA-seq, using technologies like PacBio and Oxford Nanopore, is effective for identifying novel isoforms and fusion transcripts, offering a deeper understanding of transcriptome complexity by avoiding issues with short-read limitations [7]. Small RNA-seq focuses on small non-coding RNAs, such as miRNAs, which are key regulators of gene expression. This approach is essential for investigating post-transcriptional regulation, especially in the context of disease [5]. Finally, targeted RNA-seq enables the analysis of specific subsets of genes or transcripts, making it highly suitable for clinical applications, such as biomarker discovery in precision medicine [8]. Each of these RNA-seq approaches offer distinct advantages for dissecting gene expression and uncovering novel biological phenomena, contributing to diverse fields of research and advancing our understanding of complex biological systems.

### 1.2. Advantages of RNA-Seq over Other Methods of Gene Expression Analysis

RNA-seq has revolutionized transcriptomic analyses, offering distinct advantages over traditional methods of gene expression analysis, particularly in clinical applications. Unlike microarrays, which are restricted to detecting known genes, RNA-seq can identify novel transcripts, alternative splice variants, and other RNA species, making it a more versatile tool for discovery-driven clinical research [9]. RNA-seq offers high resolution and precision, accurately determining gene boundaries, exon structures, and even single-nucleotide changes. This level of detail surpasses methods like expressed sequence tags (ESTs) or serial analysis of gene expression (SAGE), which provide a more fragmented representation of the transcriptome [10]. RNA-seq advantages regarding the identification of single nucleotide polymorphisms (SNPs) and other mutations within transcribed regions can be crucial for understanding disease mechanisms and personalizing treatments in the clinical context [11]. Microarrays are designed to detect gene-level expression and may miss complex splicing events [12]. RNA-seq can detect fusion genes, which result from the combination of two previously separate genes. Such fusions can be clinically significant, especially in the context of certain cancers. For instance, fusion gene characterization via RNA-seq was demonstrated in [13] as a useful complement in diagnosis and treatment of cancers such as acute myeloid leukemia (AML). RNA-seq can provide a wide dynamic range by detecting a vast range of expression levels, from lowly expressed to highly expressed genes. This advantage contrasts with microarrays, which can become saturated at high expression levels or fail to detect low-abundance transcripts [9]. RNA-seq also has lower background noise compared to microarrays, which can have cross-hybridization issues. This enhanced specificity is crucial in a clinical setting where accuracy is paramount [9].

Additionally, RNA-seq has a wide dynamic range, detecting expression levels from low-abundance transcripts to highly expressed genes. This feature addresses the limitations of microarrays, which may saturate at high expression levels or fail to detect lowly expressed genes [9]. RNA-seq also boasts lower background noise compared to microarrays, which are prone to cross-hybridization issues. This enhanced specificity is especially crucial in clinical settings, where accuracy is paramount for applications such as biomarker discovery and diagnostic decision-making [9]. In clinical samples infected with pathogens, RNA-seq can simultaneously assess both host and pathogen transcripts, offering insights into host-pathogen interactions and aiding in the identification of infectious agents [14]. For instance, as demonstrated in [15], researchers were able to offer insights into physiologic host-pathogen changes to host-pathogen from interaction during an infection. Additionally, RNA-seq does not rely on species-specific probes or primers, allowing for transcriptomic analysis in species that do not have commercial arrays available [9].

Finally, RNA-seq has advantages over traditional methods regarding deploying systems in clinical settings. While RT-qPCR might be more cost-effective for analyzing a small number of genes, RNA-seq has become increasingly cost-effective for large-scale studies, especially given the rapid drop in sequencing costs over the years [16]. RNA-seq has been shown to be more reproducible across labs and platforms than other techniques, which is a key attribute for clinical applications [10].

### 1.3. Tools to Extract Clinically Relevant Information from RNA-Seq Analyses

For the clinical community, especially physicians, scientists, and laboratorians who might not be deeply versed in the nuances of computational biology, the deluge of data generated by RNA-seq can be daunting. The interplay of genes, their transcription levels, and the broader context of health and disease requires an integrative approach for meaningful interpretation. Sifting through millions of sequences to derive meaningful insights demands tools that can distill complexity into an understandable format. This is where data visualization becomes invaluable. Visualization tools serve as an essential bridge in this context, converting complex data into comprehensible graphical representations and making RNA-seq data clinically actionable.

Given the widespread use and importance of RNA-seq analysis, we aimed to explore the role and impact of visualization tools in RNA-seq data analysis for clinical applications, leading to the emergence of several key research questions. Some of the key research questions we are trying to address in this paper are as follows:What are the primary visualization tools and techniques currently employed in RNA-seq data analysis, with a particular focus on clinical interpretations?How do these visualization tools enhance the understanding and interpretation of RNA-seq data within the clinical realm?What improvements can be made to existing tools, or what new tools are required, to enhance the utility and effectiveness of RNA-seq analyses in clinical applications?

To address the above research questions, this review aims to outline the current state-of-the-art data visualization techniques and tools commonly used to frame clinical inferences from RNA-seq data and point out their benefits, applications, and limitations. We intend this review to serve as a guide for improving existing visualizations and generating innovative new visualizations for physicians, laboratory scientists, and biomedical researchers interpreting RNA-seq analyses.

The review is organized according to the following sections: Section 2 (Methods) describes the methodology used to retrieve and extract articles for the systematic review, and Section 3 (Results) presents the most common data visualizations and tools used for RNA-seq analysis, examines the use of each visualization for clinical applications. Section 4 (Discussion) discusses the reviewed methods, including current challenges and future research directions. Finally, Section 5 (Conclusions) concludes the paper.

## 2. Materials and Methods

### 2.1. Search Strategy

We conducted a systematic review of English-language articles using the following online literature databases: PubMed, Scopus, Web of Science, and IEEE Xplore, adhering to the Preferred Reporting Items for Systematic Reviews and Meta-analyses (PRISMA) guidelines [17]. The workflow diagram for the systematic identification of scientific literature is shown in Figure 1.

The search terms included various combinations of keywords related to “RNA-seq” and “visualization,” connected using Boolean operators “OR” (to combine terms within the same domain) and “AND” (to link terms from different domains). A search query using the terms listed in Table 2 was used for the retrieval of primary studies. We limited our search results to review original research articles published, from 2017 to 2024, to ensure we captured the latest advancements in the field.

### 2.2. Study Selection

We started our search using the query provided in Table 2 across the databases listed in Figure 1. Based on the initial search results, we identified 126 studies. These 126 studies were saved in an excel for further examination for duplicates. We identified 35 duplicate studies in this review and removed them, leaving 91 studies for further analysis, as shown in Figure 1 in the screening section. The resulting 91 studies were further evaluated for the title/abstract screening phase. The title and abstracts of the resulting studies were first screened to identify the studies related to visualization tools and methods for RNA-seq analysis in clinical applications. This resulted in 69 eligible studies for full text review, as shown in Figure 1 (eligibility section). After identifying the 69 eligible studies for full text review, we applied additional inclusion and exclusion criteria to select the primary studies for our review (details are provided in Figure 1). Studies were eligible if they fulfilled the following inclusion criteria in our review: (1) focused on RNA-seq analysis; (2) written and published in English; (3) published between March 2017 and January 2024; (4) full text available rather than abstracts; (5) original studies published in peer-reviewed journals or appeared in conference proceedings; (6) focused on visualization tools and techniques.

Studies were not eligible if they fulfilled the following exclusion criteria in our review: (1) review articles rather than primary research; (2) developed visualization tools and techniques for other types of sequencing data; (3) full text not available; (4) published before 2017. The identified studies meeting the inclusion criteria were subsequently categorized into 4 broad categories based on the visualization tools and techniques for each type of RNA-seq data: (1) bulk RNA-seq; (2) single-cell RNA-seq; (3) circular-RNA seq; and (4) long non-coding RNA. Finally, after applying the inclusion/exclusion criteria to the 69 studies, we identified 33 studies to be included in the detailed review, as shown in Figure 1 in the included phase.

### 2.3. Data Extraction and Evaluation

With the 33 articles identified, we conducted a more detailed review of the shortlisted studies.The data were extracted from all studies meeting our inclusion criteria for the review. It consists of tables containing study information (e.g., authors’ name, title, year of study), visualization methods (e.g., volcano plots, heatmaps, backsplice visualization etc.), visualization tools developed/used, type of RNA-seq data (e.g., Bulk RNA-seq, circular RNA, single-cell RNA-seq), major outcomes, and clinically actionable insights (Table 2).

This full text review provided a deeper understanding of each visualization tool or technique, its primary applications, its merits in the clinical context, inherent limitations, and any proposed solutions or enhancements. In addition, by extracting and synthesizing this wealth of data, we aim to provide a comprehensive overview of the current landscape of RNA-seq data visualizations in clinical applications. Through this process, we aim to highlight the most widely used tools, emphasize their advantages, address the challenges, and identify potential areas for innovation and improvement. The data for all studies were independently extracted by all authors (VG, FM, and CT), and any discrepancies were resolved through mutual discussion among all authors. The extracted data were finally evaluated by all authors independently and consensus was reached through mutual agreement.

## 3. Results

We identified 126 articles in the identification phase. Thirty-five duplicate articles were removed to produce 91 articles for title and abstract screening. We further excluded 22 articles in the title and abstract screening, and accessed the full texts of the remaining 69 articles for further evaluation in the eligibility phase. Finally, 33 articles met our inclusion criteria and were considered as primary studies for this review, as shown in Table 3.

The development of visualization tools and techniques for clinical applications has grown steadily in recent years. As shown in Figure 2, studies published between 2020 and 2024 account for 73% of the total, with only nine studies published before 2020. This indicates an upward trend in the number of publications over the past five years.

Single-cell RNA-seq data is the most commonly used RNA-seq type in the reviewed studies for which visualization methods and tools have been developed in recent years. As shown in Table 3, 24 tools have been developed for single-cell RNA-seq data, while 13 tools have been developed for bulk RNA-seq data. However, only seven tools have been developed for circular RNA-seq and one tool for long non-coding RNA-seq. Different visualization methods are used in the studies to visualize RNA sequencing data that can be categorized into broader themes such as dimension reduction and clustering techniques, differential expression and gene expression visualization, RNA structure and splicing visualization, functional and network visualization, and genome browser and signal plots, as shown in Table 3. There are different types of tools, which can be broadly categorized into browser-based tools, stand-alone tools, R and Python packages, and command line tools. Based on the tools listed, the computational languages and platforms used for their development are predominantly R and Python, with some tools developed in C/C++ and web technologies like JavaScript. The major outcomes of the studies can be categorized into broader themes such as enhancing data visualization and interpretation, dimensionality reduction and clustering, identification of novel biomarkers and therapeutic targets, integration of functional and pathway analysis, facilitating single-cell RNA-seq analysis, and user accessibility and interactive exploration of sequencing data.

### 3.1. Major Themes Identified

We divided the reviewed studies into four major categories based in the RNA-seq data type: (1) bulk-RNA seq; (2) single-cell RNA-seq; (3) circular RNA-seq; and (4) long non-coding RNA. Each category comprises the visualization tools and techniques developed for clinical applications to interpret that specific RNA-seq data type. Over half of the studies (22) fell under the category of single-cell RNA-seq, with fewer studies categorized into bulk RNA-seq (nine), circular RNA-seq (seven), and long non-coding RNA (one). Since some studies covered multiple RNA-seq data types, the total count of data types mentioned (39) exceeds the number of primary studies reviewed (33).

#### 3.1.1. Single-Cell RNA-Seq

Visualizing scRNA-seq data is crucial for interpreting complex datasets, identifying distinct cell populations, understanding developmental trajectories, and uncovering underlying biological processes. Numerous tools have been developed to facilitate the visualization of scRNA-seq data, each employing various methods to represent high-dimensional data in an interpretable format.

A significant category of visualization tools focuses on dimensionality reduction techniques to project high-dimensional scRNA-seq data into lower-dimensional spaces while preserving meaningful biological relationships. For instance, Wang et al. [33] developed VASC, a deep variational autoencoder model that reduces dimensionality while addressing dropout events common in scRNA-seq data. VASC transforms high-dimensional data into two-dimensional representations, generating visual outputs like PCA, t-SNE, and heatmaps, offering clear visual distinctions of different cell populations, including rare subtypes. Similarly, Wu et al. [34] proposed SWNE (single-cell gene expression datasets with similarity-weighted non-negative embedding), which combines non-negative matrix factorization with a shared nearest neighbor network to embed both cells and genes in a two-dimensional space. SWNE retains global and local structures of the dataset, providing precise visualizations of both continuous and discrete data, and embeds genes and biological markers directly into the visualization for added context. Another tool, net-SNE, proposed by Cho et al. [47] utilized a neural network-based adaptation of t-SNE that enhances scalability and generalization. t-SNE (t-distributed stochastic neighbor embedding) is a dimensionality reduction algorithm that projects high-dimensional data into 2D or 3D, preserving local structures. Similarly, net-SNE, an extension of t-SNE, embeds graph-structured data into low-dimensional spaces by leveraging network topologies, such as edge weights and connectivity. Both algorithms are effective for visualizing complex patterns and relationships in biological or graph-based datasets. net-SNE enables mapping new cells onto existing embeddings without re-running the entire analysis, producing smooth and interpretable visualizations while preserving the local structure of the data.

Tools that integrate with existing R packages provide user-friendly interfaces for scRNA-seq data visualization. Bunis et al. [30] developed an R-based toolkit, dittoSeq, which integrates with popular analysis structures like Seurat. In the same vein, scViewer, introduced by Patil et al. [36] utilizes Seurat for data analysis and provides functionalities such as cell-type-specific gene expression visualization, co-expression analysis, and differential expression analysis across various biological conditions.

Visualization methods that preserve hierarchical relationships and capture developmental trajectories are also prominent. Garrido et al. [38] proposed DTAE, which combines clustering techniques with an autoencoder to generate visual representations of hierarchical structures in scRNA-seq data. *DTAE* preserves both global and local structures, ensuring that biologically relevant branching events are captured, making it highly applicable in developmental biology and lineage tracing studies. Topological data analysis offers another approach to visualize scRNA-seq data. Wang et al. [39] emphasized the use of Mapper, which constructs combinatorial graphs to capture topological features of high-dimensional data. Mapper’s graph-based representations preserve the continuity in the data by visualizing the continuous trajectory of cells over the space of gene expression profiles, allowing for a better understanding of cell trajectories and differentiation processes. The biological meaning of specific pathways and genes are explored through different filter functions or color coding schemes.

Some studies also employed tools that focus on comprehensive analysis pipelines, which include integrated visualization capabilities. Li et al. [43] developed scRNASequest, an end-to-end solution for scRNA-seq data analysis that includes preprocessing, harmonization, cell type annotation, and differential expression analysis. The pipeline generates interactive reports with visualization tools like UMAP plots, violin plots, volcano plots, and dot plots, facilitating seamless data management and sharing. Likewise, SCTK-QC, developed by Hong et al. [50], focuses on the generation and visualization of quality control metrics. The pipeline integrates multiple QC tasks and emphasizes visualization through detailed HTML reports, including scatterplots, violin and density plots, and dimensionality reduction plots, to visualize contamination or poor-quality cells.

#### 3.1.2. Bulk RNA-Seq

The typical workflow involves extracting total RNA from tissues or cells, preparing a complementary DNA (cDNA) library through reverse transcription and fragmentation, sequencing the cDNA using next-generation sequencing platforms to generate millions of short reads, and processing the data through quality checks and, when applicable, alignment to reference genomes or de novo assembly for organisms without reference genomes. Quantification of gene or transcript abundance follows, often normalized to account for sequencing depth and technical variability. Differential expression analysis is then applied to identify genes that are differentially expressed between experimental conditions, and functional annotation and pathway analyses help interpret the biological significance of these changes.

Visualization plays a crucial role in interpreting RNA-seq data, enabling researchers to explore complex datasets, identify patterns, and communicate findings effectively. Various tools and methods have been developed to facilitate the visualization of bulk RNA-seq data, often clustered into categories based on their approach and functionality. One category encompasses automated and user-friendly platforms that streamline RNA-seq data analysis and visualization. For instance, Cole et al. [18] proposed Searchlight, an automated platform that simplifies bulk RNA-seq exploration by generating dynamic R scripts. Searchlight automates differential expression analysis and produces standard visualizations. Users can adjust visual elements like font sizes, axis labels, and colors, which, though small adjustments, can greatly improve the clarity and customization of visual outputs for different audiences, such as clinicians, regulatory bodies, and funding agencies. Nevertheless, the main strength of these tools lies in their ability to effectively process, analyze, and visualize complex RNA-seq data.Though Searchlight is considered a strong platform, it requires some familiarity with R for advanced customization, and specialized analyses may necessitate manual adjustments, which can limit its suitability for complex projects. Similarly, BEAVR, introduced by Perampalam et al. [32], is a browser-based tool that simplifies differential gene expression analysis through a graphical user interface built with R Shiny. BEAVR uses ggplot2 to create various standard plots and pathway enrichment visualizations, aiding in the exploration of gene pathways and novel gene interactions without requiring programming expertise. While BEAVR simplifies analysis, it still requires users to format data precisely and may require some bioinformatics skills, especially for customization or advanced workflows.

Another category includes interactive visualization tools that allow real-time data exploration and customization. Bothos et al. [19] introduced SeqCVIBE, a web-based platform built on the R Shiny framework. SeqCVIBE enables users to manage and visualize RNA-seq data interactively, offering dynamic plots such as RNA signal plots, average coverage plots, faceted plots, and multidimensional scaling (MDS) for gene clustering and correlation analysis. Users can customize and export visualizations in various formats, making the tool accessible even for non-experts. Bunis et al. [30] presented dittoSeq, a universal, user-friendly toolkit for visualizing both single-cell and bulk RNA sequencing data. dittoSeq integrates with popular analysis structures like Seurat and supports various data formats. It provides a range of customizable visualizations, including dimensionality reduction plots (UMAP, PCA), scatter plots, heatmaps, bar plots, and dot plots. The outputs are compatible with ggplot2, allowing for further customization, and the tool is designed to be colorblind-friendly by default.

Interestingly, using three Bioconductor packages—pcaExplorer, ideal, and GeneTonic, Ludt et al. [31] discussed the exploration and modeling of RNA-seq data through interactive workflows. These tools offer a variety of visualization options: pcaExplorer provides PCA plots to explore gene expression patterns; ideal offers MA plots, volcano plots, and box plots for differential expression analysis; and GeneTonic integrates expression data with functional enrichment results, allowing visualization of gene signatures and pathways through interactive plots.

A different approach is seen in graph-based visualization methods, which are particularly useful for representing complex transcript structures and alternative splicing events. Nazarie et al. [20] utilized graph-based methods to capture the diversity and complexity of RNA assemblies. By representing sequencing reads as nodes and similarity scores as edges, and employing the force-directed multilevel maximally modular (FMMM) algorithm for 3D layout, this method allows for visualizations of complex transcript isoforms and splicing events. Moreover, FMMM algorithm improves the interpretability of complex biological networks by displaying clusters of biologically relevant interactions in a spatially intuitive manner. This helps in understanding the behaviour of interconnected biological components in a graph to uncover disease mechanisms. Tools like Graphia Professional play a crucial role in rendering these assembly graphs, offering a more detailed picture than traditional visualization tools. Recently, Paganin et al. [28] developed expressyouRcell, an R-based package for visualizing gene expression changes in time, space, and single cells. expressyouRcell maps gene expression data onto schematic representations of cells, creating static and dynamic cellular pictographs. Static cellular pictographs are the static images depicting the the components of a cell and cellular processes; however, dynamic cellular pictographs provide interactive visualizations of cellular processes and enable researchers to interact with the complex and evolving cellular activities. By using color-coded compartments within detailed cellular illustrations, it enables spatial visualizations of gene expression variations within cells over time, aiding in the interpretation of complex biological processes.

Dey et al. [29] explored the use of grade of membership (GoM) models for visualizing RNA-seq expression data, offering an alternative to traditional clustering methods such as k-means or hierarchical clustering. The traditional clustering methods cluster each sample into one cluster, such as a tumor sample, which can either be classified as highly expressed in one cluster or lowly expressed in other. However, GoM allows each sample to belong to multiple clusters and quantify the degree of association of a sample to a cluster. For instance, using GoM, a tumor sample can be partially highly expressed with one gene set and partially expressed with another gene set. GoM models allow samples to have partial membership in multiple clusters, reflecting the heterogeneous nature of biological data. Visualization is achieved through the structure plot, which represents the membership proportions of each sample as stacked bar charts. This method provides a nuanced view of complex data structures, revealing the degree of similarity and mixed characteristics among samples.

#### 3.1.3. Circular RNA-Seq

Circular RNAs (circRNAs) are a class of non-coding RNAs characterized by covalently closed loop structures without 5’ to 3’ polarity or polyadenylated tails. They play significant roles in gene regulation and are implicated in various diseases, including cancer. Detecting and analyzing circRNAs present unique challenges due to their circular structure and back-splicing junctions (BSJs). Several specialized tools have been developed for circRNA sequencing analysis, each employing specific visualization methods to interpret complex data.

One category of tools includes interactive web-based platforms that facilitate broad data exploration and visualization of circRNAs. Wu et al. [21] introduced CIRI-hub, a comprehensive web-based platform designed specifically for the analysis and visualization of circRNAs in cancer research. CIRI-hub integrates data from thousands of RNA-seq libraries and allows users to perform automated circRNA analyses by uploading their datasets or selecting circRNAs from the integrated database. It offers robust visualization capabilities, providing customizable plots and charts using dimension reduction techniques like PCA, MDS, t-SNE, and UMAP to distinguish between normal and tumor samples based on circRNA expression. It visualizes circRNA expression across different tissues and cancer types, allowing interactive customization of visual outputs, including color schemes and algorithms. However, some limitations of CIRI-hubinclude the need for high-performance computing resources to manage large datasets, and its effective use may require bioinformatics expertise, which can limit accessibility to non-experts. Moreover, the identification of circRNAs involves computationally intensive steps like split-read mapping and filtering false positives, which require tailored pipeline setups. This complexity is reflected in tools integrated into CIRI-hub, such as CIRIquant, which provides robust detection capabilities but demands thorough familiarity with sequencing data nuances and library preparation techniques. Another web-based tool is CircNetVis, developed by Nguyen et al. [26], which is designed to visualize and explore circRNA interaction networks. It allows users to input circRNAs in various formats and integrates multiple interaction networks to investigate relationships between circRNAs, microRNAs (miRNAs), mRNAs, and RNA-binding proteins (RBPs). Visualization is achieved through interactive network representations, where users can manipulate the layout to explore relationships within the circRNA interaction networks. Nevertheless, CircNetVis is limited to human and mouse circRNAs and relies on external databases, which can slow down real-time analysis.

Another group comprises Java-based visualization tools that provide detailed graphical representations of circRNAs and their splicing events. Zheng et al. [22] developed CIRI-vis, a command-line tool designed to visualize circRNAs and their complex splicing events. CIRI-vis integrates with other bioinformatics tools to reconstruct and quantify circRNA isoforms from RNA-seq datasets, representing the internal structure of circRNAs graphically, including forward splice events and BSJs (back-splicing junctions). This command-line interface and Java dependency may be challenging for non-experts, and it demands significant computational resources for large RNA-seq datasets. Similarly, CircView, developed by Feng et al. [24], is a Java-based desktop tool that integrates circRNA data detected from multiple tools. CircView provides an intuitive interface to explore circRNAs across different samples, displaying exons as colored arcs and introns as connecting lines, and allows users to view detailed information on exon composition and regulatory elements. Additionally, Ularcirc, introduced by Humphreys et al. [25], is an open-source software designed to visualize and analyze circRNAs by integrating both BSJs and canonical forward-splice junctions (FSJs). Ularcirc provides genomic views showing junction counts as loops overlaid on gene models, allowing users to examine exon–intron boundaries and internal splicing patterns within circRNAs. CircView may struggle with performance on large datasets and requires an understanding of circRNA biology, while Ularcirc needs high sequencing depth and may lack efficiency in filtering false positives.

Specialized tools focusing on specific aspects of circRNA analysis include INTEGRATE-Circ and INTEGRATE-Vis, developed by authors in [23]. These tools specialize in detecting and visualizing fusion-derived circRNAs (fcircRNAs) formed through backsplicing events within fusion gene transcripts by integrating RNA and whole-genome sequencing data. Both need substantial computational power and rely on whole-genome sequencing data for accurate fcircRNA detection, with challenges in distinguishing between true fusion events and read-through events. Using both tools INTEGRATE-Circ and INTEGRATE-Vis in tandem enables seamless integration from detection to visualization. However, INTEGRATE-Circ can function independently for detection purposes, and INTEGRATE-Vis can visualize outputs from compatible tools. For best results, combining both ensures comprehensive analysis and user-friendly presentation. Lin et al. [27] introduced CircVIS, a web-based platform that annotates circRNAs based on their subcellular localizations and aligns them to reference transcripts, comparing derived open reading frames (ORFs) to their parental proteins. The visualization integrates circRNA annotations with reference transcripts and exon usage, aiding in understanding how circRNAs function across different cellular compartments. CircVIS has some challenges because of its limited availability of compatible datasets and a scarcity of subcellular RNA-seq data, necessitating additional experimental validation for certain circRNAs.

These specialized tools and their visualization methods empower researchers to unravel the complexities of circRNAs. Interactive web-based platforms like CIRI-hub and CircNetVis facilitate broad data exploration and interaction networks, while Java-based tools like CIRI-vis and CircView provide detailed graphical representations of circRNA structures and splicing events. Tools like Ularcirc and INTEGRATE-Vis offer specialized visualization of splicing patterns and fusion-derived circRNAs, contributing to a deeper understanding of circRNA biology. CircVIS adds another dimension by focusing on subcellular localization and functional annotation visualization, enabling researchers to predict circRNA behavior based on cellular compartments. Collectively, these tools advance our knowledge of circRNAs’ roles in gene regulation and disease.

#### 3.1.4. Long Non-Coding RNA-Seq

Long non-coding RNAs (lncRNAs) are RNA molecules longer than 200 nucleotides that do not code for proteins but play crucial roles in regulating gene expression and various biological processes. Sequencing and analyzing lncRNAs present unique challenges due to their low expression levels, tissue specificity, and complex regulatory functions. To address these challenges, specialized tools and software have been developed to enhance lncRNA analysis. One key platform is SeqCVIBE, as highlighted in the paper [19]. SeqCVIBEis an R Shiny web application that allows researchers to perform differential expression analysis, calculate RNA abundances over both annotated and non-annotated genomic regions, and visualize results in real-time. This capability is particularly beneficial for investigating novel lncRNAs, such as WiNTRLINC1, which is involved in Wnt signaling in cancer studies. The platform offers a database of pre-analyzed RNA-seq data, facilitating comprehensive lncRNA research. The primary purpose of SeqCVIBE seems to be providing a comprehensive platform for managing, visualizing, and analyzing RNA-seq data, with features such as dynamic signal plotting, differential gene expression analysis (DGEA), and genome browsing. It was designed to facilitate RNA-seq workflows broadly rather than being developed specifically for lncRNA analysis. SeqCVIBE’s ability to calculate RNA abundances and visualize data in non-annotated genomic regions makes it particularly suitable for studying novel or uncharacterized lncRNAs.

Another significant tool is PANDORA, a weighted p-value combination algorithm designed to optimize differential gene expression analysis (DGEA) [19]. PANDORA excels in lncRNA-specific settings due to its ability to address key RNA-seq biases like gene length and normalization framework variability. Its simulation-based weighted p-value scheme optimizes the precision-sensitivity tradeoff, making it particularly effective for low-count RNA molecules such as lncRNAs. PANDORA is useful for analyzing lncRNAs by controlling biases from different normalization frameworks and is implemented in Bioconductor packages like metaseqR and metaseqR2.

Visualization is critical in lncRNA analysis for interpreting complex RNA-seq data. SeqCVIBE excels in providing visualization methods tailored for lncRNA research. It generates dynamic RNA-seq signal plots, allowing users to visualize averaged RNA-seq signal coverage for both known and novel lncRNAs across samples, including non-annotated regions where new transcription events are detected. An essential feature within SeqCVIBE is the integration of JBrowse, a genome browser enabling detailed mapping of RNA-seq signals across the genome. This integration helps visualize lncRNA interactions with other genomic elements and their potential regulatory roles. The application also supports statistical visualization methods like principal component analysis (PCA) and multi-dimensional scaling (MDS) plots to assess sample variability and relationships between lncRNAs and other genes. JBrowse is suitable for lncRNA visualization due to its ability to handle sparse expression data and provide detailed views of genomic regions. Its interactive zoom and multi-track alignment help identify subtle signals and regulatory components like enhancers or splice sites. Real-time customization allows researchers to explore novel lncRNAs and their genomic contexts effectively. Tangible examples, such as discovering lncRNAs with low read coverage or mapping enhancer–promoter interactions, would enhance its relevance to lncRNA research. Beyond SeqCVIBE, visualization methods for lncRNA sequencing often involve heatmaps, volcano plots, and gene expression profiles to represent differential expression patterns. Tools like the Integrative Genomics Viewer (IGV) are utilized for detailed views of genomic alignments and annotations, aiding in the identification of novel lncRNAs and understanding their regulatory roles. The integration of these advanced tools and visualization methods significantly enhances the understanding of lncRNA functionality and their roles in various biological processes, as the paper emphasizes, ultimately contributing to advancements in fields such as cancer research and functional genomics.

## 4. Discussion

RNA-seq has the potential to transform clinical diagnostics, especially in oncology, by offering a powerful way to identify disease-specific biomarkers. This ability allows diagnostic laboratories to gain valuable insights into diseases like cancer. For example, RNA-seq excels at identifying differentially expressed genes (DEGs) through processes such as alternative splicing, which plays a critical role in cancer and tumor classification [51]. DEGs are critical to RNA-seq investigations as they provide insights into changes in gene expression under different experimental conditions and time points. These molecular assays enable earlier and more accurate disease diagnoses, ultimately improving patient outcomes through timely and targeted interventions. RNA-seq also plays a pivotal role in understanding disease mechanisms. By revealing regulatory pathways and transcriptomic patterns, laboratories can uncover actionable targets for precision therapies. For instance, RNA-seq has identified disease phenotypes linked to dysregulated gene networks, enabling therapies that address root causes rather than just symptoms. Traditional methods for analyzing gene expression like RT-qPCR, microarrays, and Northern blotting still have their merits and specific applications. Indeed, RT-qPCR remains the gold standard for targeted gene expression studies, primarily due to its cost-effectiveness. We align with other studies [52] in suggesting that RNA-seq is expected to become more cost-effective as it sees wider adoption. Its reproducibility and reliability are essential for clinical applications, where data cross-validation and compliance with regulatory standards are paramount.

The systematic review conducted in this study aimed to explore the current landscape of visualization techniques and tools used in RNA-seq data analysis, with a particular focus on their applications and limitations in clinical contexts. By analyzing 33 primary studies published between 2017 and 2024, we have provided a comprehensive overview of state-of-the-art visualization methods employed across different types of RNA-seq data, including bulk RNA-seq, single-cell RNA-seq, circular RNA-seq, and long non-coding RNA-seq. Our findings revealed a significant emphasis on single-cell RNA-seq data, with 24 out of 33 tools developed for this data type. This trend reflects the growing importance and adoption of single-cell technologies in clinical research, as they offer unparalleled resolution in understanding cellular heterogeneity and disease mechanisms at the individual cell level. Emerging technologies like single-cell RNA-seq and spatial transcriptomics are further enhancing our understanding of disease pathology and treatment responses. Single-cell RNA-seq offers unprecedented resolution, capturing cellular heterogeneity and differentiation patterns. This is particularly valuable in cancer, where identifying rare cell subpopulations or stem cell-like properties can inform treatment strategies. Spatial transcriptomics adds another layer, linking gene expression to tissue architecture—a transformative advance for pathology. While these technologies hold great promise, they also present challenges. For instance, interpreting high-dimensional data from single-cell RNA-seq requires sophisticated tools like UMAP or t-SNE plots, which reduce complexity into intuitive 2D visualizations. These tools highlight rare cell types or disease-associated subpopulations. Similarly, trajectory analyses, often visualized as tree-like structures, map cellular differentiation paths, providing insights into processes like cancer progression or immune dysfunction. However, correlating single-cell findings to bulk tissue data and standardizing protocols for emerging techniques like spatial transcriptomics remain key obstacles.

Previous studies have highlighted the transformative impact of single-cell RNA-seq in fields like oncology and immunology, enabling the discovery of rare cell populations and elucidating complex biological processes [33,40]. Our review corroborates these trends, demonstrating that visualization tools are keeping pace with advancements in single-cell sequencing technologies.

In contrast, fewer tools were developed for bulk RNA-seq (13 tools), circular RNA-seq (7 tools), and long non-coding RNA-seq (1 tool). This discrepancy may be attributed to the complex nature of single-cell data requiring more sophisticated visualization approaches, as well as the burgeoning interest in single-cell analyses in clinical applications. However, bulk RNA-seq remains a staple in clinical research for its utility in profiling average gene expression across cell populations. Tools like Searchlight [18] and BEAVR [32] streamline differential expression analysis and visualization for bulk RNA-seq data, facilitating efficient interpretation of gene expression changes relevant to disease states.

Visualization methods identified in the studies include dimensionality reduction techniques such as t-SNE and UMAP, clustering algorithms, heatmaps, and network visualizations. These methods are crucial for interpreting high-dimensional RNA-seq data and uncovering underlying patterns and structures. For instance, tools like SWNE [34] and VASC [33] enhance the visualization of single-cell data by preserving both global and local data structures, which is essential for accurate interpretation of cellular differentiation pathways and disease progression. Visualizations like heatmaps and volcano plots are indispensable for interpreting RNA-seq data in this context. Heatmaps provide an intuitive way to group patient samples based on shared expression patterns, aiding both disease classification and biomarker discovery. Meanwhile, volcano plots help pinpoint genes with high diagnostic potential by showcasing statistical significance against fold changes. For instance, these plots can identify circRNAs with regulatory roles in tumor progression and metastasis [21]. Such tools make it easier for clinical laboratories to connect molecular findings to real-world diagnostics.

Visualizations such as pathway enrichment plots, Sankey diagrams, and gene interaction networks are also instrumental. Sankey diagrams are used for flow analysis. They consists of nodes and flow arrows. Nodes represent the entities and arrows represent the flow with the width of arrow is proportional to the value it represent. For RNA-seq analysis, Sankey diagrams are paramount to visualize DEGs, alternative splicing events, and pathway enrichment. For instance, to compare two disease conditions, Sankey diagrams can provide information on the gene expression and its magnitude through the width of arrows.

Pathway enrichment plots are used to identify biological pathways significantly associated with a set of genes in RNA-seq studies. Pathway enrichment plots highlight overrepresented pathways in DEGs, aiding in the identification of molecular drivers of disease. Sankey diagrams trace gene flows through regulatory networks, while interaction networks offer a dynamic view of transcriptomic changes, uncovering potential therapeutic targets. Interestingly, Sankey diagrams, traditionally used in research, are finding clinical applications. For example, they have been adapted to track cancer symptom trajectories, providing insights for symptom monitoring [53]. Similarly, pathway enrichment plots can shed light on immune-related pathways in autoimmune disorders, paving the way for innovative diagnostic assays. However, bridging the gap between identified molecular mechanisms and actionable treatments requires interdisciplinary collaboration and extensive validation. Developing tools that balance simplicity and accuracy will be critical for integrating these visualizations into clinical workflows.

Our second research question aimed to understand how these visualization tools enhance the understanding and interpretation of RNA-seq data within the clinical realm. The tools identified in our review offer various benefits, such as enabling the identification of differentially expressed genes, uncovering novel biomarkers, and visualizing complex gene expression patterns associated with diseases. For example, CIRI-hub [21] specializes in identifying novel circular RNAs with diagnostic potential in cancers, while expressyouRcell [28] maps gene expression changes to cellular compartments, aiding in disease modeling. These tools play a pivotal role in translating complex RNA-seq data into clinically actionable insights. They allow clinicians and researchers to interpret large datasets intuitively, facilitating the discovery of disease-associated genes, pathways, and cell populations. This is particularly important in personalized medicine, where understanding individual variability at the molecular level is crucial for tailored treatments [54].

The predominance of visualization tools for single-cell RNA-seq data underscores a significant shift in biomedical research toward single-cell analyses. This shift has profound implications for clinical applications, as it enables a more nuanced understanding of diseases at the cellular level. The ability to identify rare cell types and understand cellular heterogeneity can lead to the development of more precise diagnostic tools and therapeutic strategies [42,43]. Moreover, the increasing number of visualization tools indicates a recognition of the importance of making complex RNA-seq data accessible to a broader range of users, including clinicians who may not have extensive computational expertise. Tools with user-friendly interfaces, such as dittoSeq [30] and scViewer [36], lower the barriers to entry, allowing more researchers to leverage RNA-seq data in clinical studies. Although both dittoSeq and scViewer have been widely utilized in research settings for analyzing and visualizing RNA-seq data, specific examples of their adoption in clinical laboratories remain sparse. We highlight these tools as models to illustrate how user-friendly interfaces can potentially bridge the gap between complex data analysis and clinical decision-making. There are other popular tools, like Galaxy [55], which are well-known among biomedical researchers for their flexibility and ease of use but have yet to see widespread adoption in clinical settings. Addressing barriers to implementation, such as integration with clinical workflows and regulatory compliance, could pave the way for broader acceptance of these tools in clinical applications. However, the relatively limited number of tools for circular RNA-seq and long non-coding RNA-seq suggests potential gaps in the field. Given the emerging roles of these RNA types in gene regulation and disease, there is a pressing need to develop more specialized visualization tools to facilitate their study [23,27]. This could unlock new avenues for understanding disease mechanisms and identifying novel therapeutic targets.

Despite these advancements, transitioning RNA-seq findings into practical clinical assays remains challenging. Validating biomarkers across diverse patient populations requires large-scale datasets, which create logistical and computational bottlenecks for many laboratories. For instance, obtaining sufficient patient samples representative of different age groups, ethnic backgrounds, and disease stages often involves multicenter collaborations that are challenging to coordinate. Moreover, integrating new biomarkers into existing workflows without disrupting established practices requires careful planning and validation. For example, introducing RNA-seq-derived biomarkers into clinical workflows may necessitate standardization of RNA extraction and library preparation protocols, which vary widely across laboratories. Furthermore, ensuring compatibility with existing diagnostic platforms and regulatory compliance involves extensive optimization and rigorous testing, often leading to delays in clinical adoption. Overcoming these barriers is essential to fully harness RNA-seq’s potential in diagnostics. Indeed, in oncology, RNA-seq has become a cornerstone for molecular diagnostics and personalized medicine. By tailoring treatments to a patient’s molecular profile, RNA-seq can improve treatment efficacy while minimizing side effects. Kaplan–Meier survival plots, a popular RNA-seq visualization method in oncology, offer insights into the relationship between gene expression and survival outcomes [56]. These plots help identify prognostic biomarkers that inform clinical decisions. Commercial tools have already started incorporating survival plots to guide diagnostics, but their adoption faces hurdles like a lack of in-house bioinformatics expertise and the need for robust clinical validation frameworks. Additionally, personalized approaches require sophisticated algorithms to integrate multi-omics data, addressing challenges like intratumoral variability and therapeutic resistance.

Gene expression profiling using RNA-seq provides a comprehensive snapshot of cellular processes, offering valuable insights into disease progression and treatment responses. Custom gene expression panels tailored to individual patients enable highly targeted therapies. Visualizations like bar plots, PCA plots, and violin plots enhance the interpretability of these data. For instance, PCA plots group samples by similar expression profiles, revealing patterns related to disease progression, while violin plots highlight gene expression variability that might influence therapy responses. Despite their utility, these approaches face challenges such as variability across sequencing platforms and difficulties with data reproducibility. Normalizing data to mitigate batch effects is a persistent issue, as inconsistencies can compromise downstream analyses. Advanced computational tools and expert interpretation are crucial to overcome these challenges.

Maintaining data quality is fundamental to ensuring reproducibility and reliability in RNA-seq applications. Clinical laboratories employ rigorous quality control protocols to verify data integrity, filtering out low-quality samples that could skew results. Visualizations like box plots, PCA plots for batch effect assessment, and quality heatmaps are invaluable for assessing data reliability. However, the lack of universal quality control standards across sequencing methods complicates efforts to standardize analyses. Additionally, computational tools must meet clinical-grade requirements to ensure accurate and consistent results.

Currently, many RNA-seq visualization tools cater to research settings and are not optimized for clinical decision-making. To bridge this gap, some tools, like the FDA-approved Oncomine (Thermo Fisher Scientific) [57] and cBioPortal [58], incorporate visualizations like bar plots and Kaplan–Meier curves into user-friendly interfaces. Kaplan–Meier curves are particularly valuable in cancer research, where RNA-seq data can identify gene expression signatures associated with patient survival. For example, genes overexpressed in specific cancers, as detected by RNA-seq, can be categorized into high- and low-expression groups, and Kaplan–Meier plots can illustrate differences in survival rates between these groups. Despite their utility, these tools often fall short of integrating RNA-seq analysis with electronic health records or clinical decision support systems, limiting their adoption in routine clinical workflows. Integrating RNA-seq data with electronic health records has the potential to improve clinical workflows, with improved disease diagnosis, prognosis, and understanding of treatment effects, by combining clinical context with genetic insights. This will help in advancing precision medicine and generate novel insights on various complex disease conditions. Further research should focus on developing pipelines that automate the integration of RNA-Seq data into electronic health records, accompanied by visualization tools tailored to clinical needs. Examples include dashboards for monitoring disease progression through gene expression trends or predictive algorithms that flag potential complications based on transcriptomic patterns. Addressing these challenges will bring us closer to realizing the full potential of RNA-Seq in enhancing patient care.

While our review provides valuable insights, it is essential to acknowledge certain limitations. The studies included were limited to those published in English and available in full text, potentially excluding relevant research in other languages or inaccessible publications. A key limitation of this review is that we relied on tool descriptions provided in the literature or documentation and did not independently test all the tools due to time and resource constraints. Future efforts should prioritize direct testing of tools to provide more accurate assessments of their usability and applicability. Additionally, future research should focus on developing more intuitive and accessible visualization tools that cater to users with varying levels of computational expertise. Integrating graphical user interfaces and providing comprehensive documentation can enhance usability for clinicians and researchers without programming backgrounds.

## 5. Conclusions and Future Research

In conclusion, the findings emphasize the need for the continued development of specialized, user-friendly visualization methods that can handle diverse RNA-seq data types. By addressing current limitations and focusing on future research directions, we can enhance the utility and effectiveness of RNA-seq analyses in clinical applications, ultimately contributing to improved outcomes in healthcare.

There is also a need to expand visualization tools to underrepresented RNA-seq data types. Given the limited tools for circular RNA-seq and long non-coding RNA-seq, future efforts should prioritize developing visualization methods for these data types. This could involve adapting existing tools or creating new ones that address the unique challenges associated with these RNA species. Combining RNA-seq data with other omics data, such as proteomics and epigenomics, can provide a more holistic view of biological systems. Developing visualization tools capable of integrating and representing multi-omics datasets can enhance our understanding of complex disease processes [26,38]. As datasets continue to grow in size and complexity, tools must be optimized for performance and scalability. Leveraging cloud computing and advanced algorithms can improve the efficiency of data processing and visualization [48,49]. Establishing standards for data formats and visualization practices can facilitate data sharing and collaboration. Tools that support interoperability between different platforms and data types will be invaluable for large-scale collaborative research efforts. Moreover, future research should focus on validating these tools in clinical settings and assessing their impact on diagnosis, prognosis, and treatment decisions. Looking ahead, advancements in RNA-seq visualizations must focus on usability for clinicians. Integrating artificial intelligence and automation can help generate simplified, interpretable outputs. For instance, incorporating deep learning into spatial transcriptomics workflows could overlay transcriptomic data onto digital pathology slides, making molecular findings accessible within routine clinical workflows. Such innovations will be pivotal in translating RNA-seq’s promise into actionable clinical solutions.

## Figures and Tables

**Figure 1 bioengineering-12-00056-f001:**
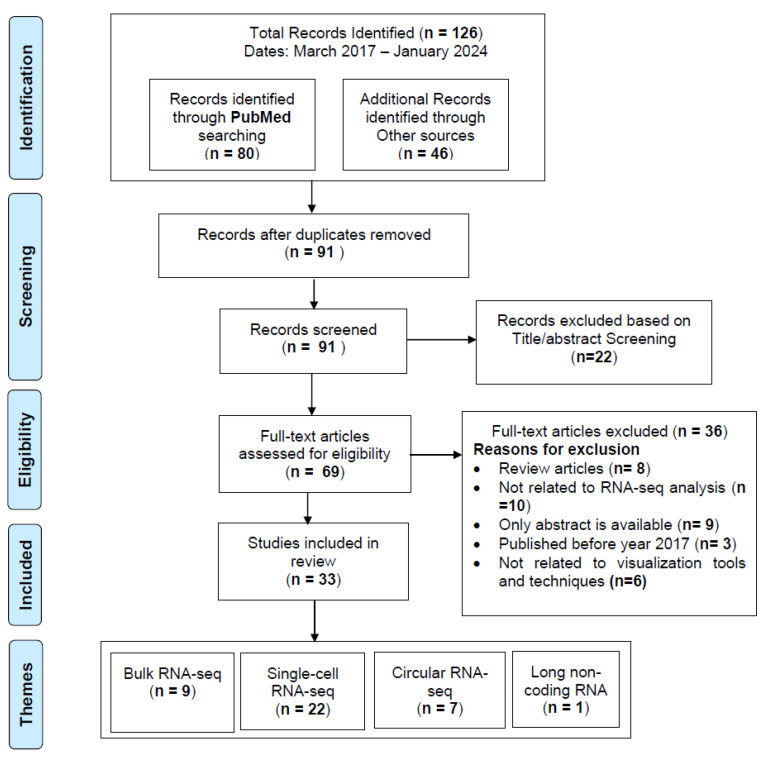
PRISMA workflow for systematic identification of scientific literature.

**Figure 2 bioengineering-12-00056-f002:**
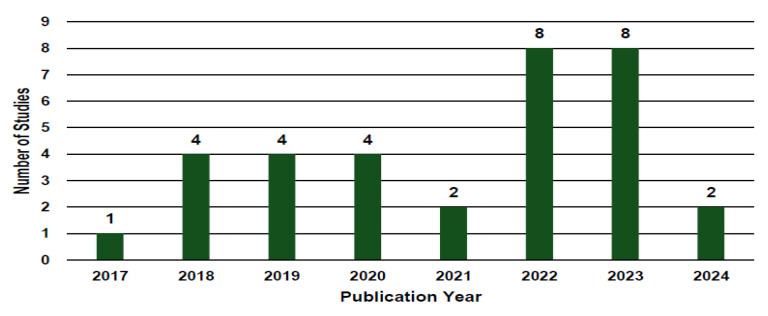
Distribution of studies by publication year.

**Table 1 bioengineering-12-00056-t001:** Different types of RNA-seq.

A. Bulk RNA Sequencing	
Total RNA Sequencing (coding RNA and non-coding RNA):	Sequencing entire transcriptome, both coding and noncoding RNA (depleting over abundant rRNA) ideal for studies seeking a comprehensive, broader picture of cellular processes.
mRNA Sequencing (coding RNA):	Sequencing only coding RNA (enriched by selectively captured mRNA) ideal for studies centered on gene expression and regulation. Can identify known and novel isoforms in the coding transcriptome, detect gene fusions, and measure allele-specific expression.
**B. Specialized RNA Sequencing**	
Single-Cell RNA-Seq (cellular heterogeneity):	Sequencing (typically just coding RNA) on individually isolated cells to examine distinct gene expression profiles to get a high-resolution view of cellular heterogeneity and deeper picture into cellular functions and states.
Spatial Transcriptomics (spatial context):	Sequencing (typically just coding RNA) which maps gene expression within the spatial architecture of tissues, preserving the positional information of transcripts. Enables visualization of where specific genes are expressed within the tissue.
**C. Application-Specific Enrichment Strategies**	
Small RNA Sequencing (regulatory RNAs):	Specialized sequencing designed to specifically evaluate small RNA molecules in a sample. May include microRNAs (miRNAs), small interfering RNAs (siRNAs), piwi-interacting RNAs (piRNAs), and other small non-coding RNAs (sncRNAs).
RNA Exome Capture Sequencing (coding regions):	Sequencing approach designed to evaluate the exonic regions of the transcriptome (while excluding the non-coding intrinsic elements).
Targeted RNA Sequencing (specific transcripts):	Sequencing approach designed to evaluate selective transcripts or regions of interest, such as genes, exons, fusion transcripts, or other target RNA molecules.
Ultra-Low-Input RNA Sequencing (minimal samples):	Sequencing approach designed to evaluate RNA from extremely small amounts of starting material. Ideal for scenarios where sample is limited (such as rare cell populations or from biopsies).

**Table 2 bioengineering-12-00056-t002:** Search query for the study retrieval.

((RNA-seq OR RNA sequencing) AND (analysis OR data analysis) AND (visualization OR plot OR chart OR graph OR diagram) AND (clinical OR medical) AND (application))

**Table 3 bioengineering-12-00056-t003:** A brief summary of reviewed studies.

References	Visualization Methods	Tools	RNA-Seq Data Types	Major Outcome Identified	Clinically Actionable Insights
Cole et al., 2021 [18]	MA plots, heatmaps, volcano plots, PCA plots	Searchlight with R scripts (1 tool)	Bulk RNA-seq	Streamlines differential expression analysis and visualization	Efficient analysis for disease-related gene expression
Bothos et al., 2022 [19]	RNA signal plots, MDS, PCA, heatmaps	SeqCVIBE, JBrowse (1 tool)	Bulk RNA-seq, Long non-coding RNAs	Facilitates real-time RNA-seq data exploration and analysis	Discovery of novel RNAs like lncRNAs for disease research
Nazarie et al., 2019 [20]	Graph-based 3D RNA-seq assembly graphs	Graphia Professional	Bulk RNA-seq	Captures complexity of transcript isoforms and splicing events	Identifies transcript diversity important for disease mechanisms
Wu et al., 2023 [21]	PCA, MDS, t-SNE, UMAP	CIRI-hub	Bulk RNA-seq, Circular RNAs	Identifies novel circRNAs with diagnostic potential in cancers	Potential diagnostic biomarkers for cancer
Zheng et al., 2020 [22]	Circular RNA splice events visualization	CIRI-vis	Circular RNA-seq	Reconstructs and visualizes circRNA isoforms and splicing events	Enhances understanding of circRNA role in diseases
Webster et al., 2023 [23]	Fusion-derived circular RNA visualization	INTEGRATE-Circ & INTEGRATE-Vis	Circular RNA-seq	Sensitive detection of fusion-derived circular RNAs	Potential for discovering cancer biomarkers
Feng et al., 2019 [24]	Circular RNA structures, exons, and introns	CircView	Circular RNA-seq	Facilitates exploration of circRNAs with exon composition and regulatory elements	Identifies circRNAs involved in cancer progression
Humphreys et al., 2019 [25]	Sushi genomic visualization, backsplice junction visualization	Ularcirc	Circular RNA-seq	Visualizes circRNA biogenesis, open reading frames, and splicing events	Provides insights into circRNA function and regulation in diseases
Nguyen et al., 2024 [26]	Interaction network visualization	CircNetVis	Circular RNA-seq	Investigates circRNA interactions with miRNAs and mRNAs	Aids in studying circRNA roles in gene regulation
Lin et al., 2022 [27]	Functional annotation of circRNAs	CircVIS	Circular RNA-seq	Visualizes circRNA isoforms and subcellular localizations	Studies circRNA roles in gene regulation
Paganin et al., 2023 [28]	Gene expression changes in subcellular compartments	express youRcell	Bulk RNA-seq, Single-cell RNA-seq	Maps gene expression changes to cellular compartments	Visualizes dynamic changes for disease modeling
Dey et al., 2017 [29]	Structure plot for sample clustering	CountClust	Bulk RNA-seq, Single-cell RNA-seq	Identifies mixed memberships of samples in multiple clusters	Uncovers gene expression heterogeneity in disease samples
Bunis et al., 2020 [30]	Dimensionality reduction plots, heatmaps, scatter plots	dittoSeq	Bulk RNA-seq, Single-cell RNA-seq	Universal toolkit for RNA-seq data visualization	Identifies differentially expressed genes in clinical studies
Ludt et al., 2022 [31]	PCA plots, MA plots, volcano plots, heatmaps	pcaExplorer, ideal, GeneTonic (3 tools)	Bulk RNA-seq, Single-cell RNA-seq	Integrates functional enrichment with gene expression data	Identifies key pathways and gene signatures in diseases
Perampalam et al., 2020 [32]	PCA, heatmaps, volcano plots	BEAVR	Bulk RNA-seq	Simplifies DGE analysis and visualization	Applicable in oncology for diagnostics and therapy
Wang et al., 2018 [33]	PCA, t-SNE, heatmaps	VASC	Single-cell RNA-seq	Reduces scRNA-seq data dimensionality while handling dropout events	Identifies rare cell subtypes critical in disease research
Wu et al., 2018 [34]	2D embedding of cells and genes	SWNE	Single-cell RNA-seq	Captures global and local structures of cell states	Reveals cell differentiation trajectories important for developmental biology
Lewsey et al., 2022 [35]	Cluster and values mode, 3D UMAP clustering	scCloudMine	Single-cell RNA-seq	Allows visualization of scRNA-seq data for comparative studies	Enables discovery of complex biological processes
Patil et al., 2023 [36]	UMAP, feature plots, violin plots	scViewer	Single-cell RNA-seq	Provides co-expression analysis and differential expression insights	Identifies disease-specific cell type patterns
Liu et al., 2020 [37]	ssPCA plots	ssPCA	Single-cell RNA-seq	Balances local and global structure in scRNA-seq data	Captures cell progression and transitions for disease insights
Garrido et al., 2022 [38]	Nonlinear autoencoder-based 2D embedding	DTAE	Single-cell RNA-seq	Visualizes hierarchical structures in scRNA-seq data	Tracks cell differentiation in disease contexts
Wang et al., 2018 [39]	Mapper graphs	Mapper	Single-cell RNA-seq	Preserves local and global structures of scRNA-seq data	Identifies branching trajectories in cell differentiation
Linderman et al., 2019 [40]	Accelerated t-SNE and heatmap visualization	FIt-SNE	Single-cell RNA-seq	Visualizes gene expression patterns and cell clusters in large datasets	Identifies complex gene expression patterns in diseases
Yousuff et al., 2024 [41]	2D scatter plots	CP-PaCMAP	Single-cell RNA-seq	Enhances compactness and structure preservation in dimensionality reduction	Useful in classifying or clustering cells based on gene expression
Hoek et al., 2021 [42]	t-SNE, UMAP, violin plots, heatmaps	WASP	Single-cell RNA-seq	Enables interactive data exploration with minimal computational expertise	Applicable in studying tumor heterogeneity and cellular differentiation
Li et al., 2023 [43]	UMAP, violin plots, volcano plots	scRNASequest	Single-cell RNA-seq	Harmonizes scRNA-seq data across large datasets	Identifies cell-type-specific gene expression
Vasighizaker et al., 2022 [44]	t-SNE, PCA, Isomap	MLLE, ICA	Single-cell RNA-seq	Improves dimensionality reduction and clustering for novel cell type discovery	Identifies biologically relevant cell subtypes and pathways
Hsu et al., 2023 [45]	CA biplots, UMAP	corral	Single-cell RNA-seq	Improves clustering accuracy and batch integration	Identifies cell subpopulations and gene associations
Xu et al., 2023 [46]	Euclidean and hyperbolic space visualizations	DV framework	Single-cell RNA-seq	Preserves data structures while correcting for batch effects	Uncovers cellular hierarchies and disease-related gene networks
Cho et al., 2018 [47]	Neural t-SNE embeddings	net-SNE	Single-cell RNA-seq	Scales to large datasets with parametric approach	Generalizable to new cells without re-analysis
Hou et al., 2022 [48]	Grid-based gene expression visualization	SCUBI	Single-cell RNA-seq	Provides unbiased, scalable visualization of scRNA-seq data	Helps in exploring gene expression in disease studies
Xu et al., 2023 [49]	UMAP with graph embeddings	autoCell	Single-cell RNA-seq	Handles sparse scRNA-seq data for trajectory inference	Denoises data for identifying cellular subpopulations
Hong et al., 2022 [50]	Violin plots, density plots, scatterplots	SCTK-QC	Single-cell RNA-seq	Provides comprehensive quality control for scRNA-seq data	Ensures high-quality data for accurate downstream analysis

## Data Availability

Data are contained within the article and Appendix A.

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
