# Peer review of "Exploring RNA-Seq Data Analysis Through Visualization Techniques and Tools: A Systematic Review of Opportunities and Limitations for Clinical Applications"

_bioengineering, 2025, doi:10.3390/bioengineering12010056_

Round 1

Reviewer 1 Report

Comments and Suggestions for Authors

Manzoor et al. present a comprehensive review on RNAseq visualization tools. They perform a literature research and receive, after filtering, 33 papers, which are discussed. The majority of the papers deals with scRNAseq, but there are also papers on bulk, circular and lncRNAs. The review is a pleasure to read and gives a valuable overview over the field. Since visualization is an important part of RNAseq analysis, it will be of interest to the readers.

Author Response

General Comments: Manzoor et al. present a comprehensive review on RNAseq visualization tools. They perform a literature research and receive, after filtering, 33 papers, which are discussed. The majority of the papers deals with scRNAseq, but there are also papers on bulk, circular and lncRNAs. The review is a pleasure to read and gives a valuable overview over the field. Since visualization is an important part of RNAseq analysis, it will be of interest to the readers.

Response: We thank the reviewer for sharing a good feedback of our manuscript.

Reviewer 2 Report

Comments and Suggestions for Authors

This review emphasizes the benefits of RNA sequencing (RNA-seq) in clinical research and patient care, as well as its revolutionary role in transcriptome analysis. RNA-seq is a potent technique that uses next-generation sequencing to assess gene expression in a comprehensive way. This allows for the study of both coding and non-coding RNA as well as phenomena like mutations, gene fusions, and alternative splicing. 126 articles (2017–2024) are thoroughly reviewed, and the most common visualization tools for clinical RNA-seq analysis are identified. Of them, 56% concentrate on single-cell data, 23% on bulk RNA-seq, and lesser percentages on circular and long non-coding RNA.

Clinical professionals and researchers can better understand the complicated data produced by RNA-seq by using visualization tools. The study describes techniques for bulk, single-cell, and targeted RNA-seq, highlighting its uses in personalized medicine and diagnostics. It also points out gaps, recommending both the creation of new tools and enhancements to current ones in order to satisfy clinical needs. In general, the review provides guidance on how to use RNA-seq and visualization methods to improve knowledge of gene expression in both health and illness.

Although the evaluation of RNA sequencing (RNA-seq) data visualization for clinical applications is well-structured, there are a few potential errors and areas for improvement.

In the introduction, "host-pathogen interactions" and "fusion genes" are discussed, but no clinical examples are provided, which would increase their applicability.
The PRISMA recommendations are cited in the abstract and methods section, although the text does not go into great depth about how the inclusion and exclusion criteria were used. How were duplicates managed? What were the study quality criteria?
Only Web of Science, IEEE Xplore, PubMed, and Scopus were searched. Why not incorporate clinical trial registries (like ClinicalTrials.gov) or biomedical conference proceedings, considering the emphasis on clinical applications?
Even if RNA-seq is marketed as being better, issues where more traditional techniques (such RT-qPCR or microarrays) are still preferred (like cost-effectiveness for small-scale investigations) should be covered in the discussion.

Transcript dropouts in scRNA-seq can result in under-representation of lowly expressed genes, which is an issue important addressing. In contrast, RNA-seq's "wide dynamic range" does detect both low and high expression levels.

Table 2's Boolean operators and phrases are too general and may return research that aren't relevant. For example, "visualization OR plot OR chart OR graph OR diagram" may result in a large number of unrelated or non-clinical publications.

The claim that each author assessed the retrieved data "independently" runs counter to the prior reference to mutual agreement. It's unclear how independence and reaching an agreement can coexist.

According to the text, 91 articles remain after 126 articles were first found and 35 duplicates were eliminated. The flow described later (69 articles reviewed after exclusion) is not consistent with this, though. It is necessary to clarify whether title/abstract screening exclusions took place prior to deduplication.

Many tools (such as "identifies gene expression patterns" or "enhances understanding") are mentioned with overlapping results. Redundancy may be decreased by classifying tools according to their functions. Additionally, some tools like VASC and SWNE have thorough descriptions, while others like net-SNE have only a quick description. Clarity and balance would be enhanced by uniformity in detail.

While CIRI-hub's description promotes its advanced features, it does not explicitly mention weaknesses such as its computational demands, as strongly as it does highlight its benefits. "It requires high-performance computing resources for handling large datasets, and effective use may need bioinformatics expertise, limiting accessibility for non-experts." Examples or certain metrics could be used to further elucidate accessibility concerns.

CircNetVis's support for human and mouse circRNAs is mentioned in the discussion, but it is not made clear whether upgrades or extensions for other organisms are anticipated or feasible, which may limit its applicability in more general research settings. CircNetVis's use of the phrase "interaction networks" is vague about what these networks are. Do these networks consist solely of associations, regulatory networks, or molecular interaction networks?

Although the section includes a list of tools such as Ularcirc, CircView, and CIRI-vis, it does not directly examine their advantages and disadvantages. Which particular use cases are specific to each tool? Do some tools perform better in exploratory research, while others are better suited for clinical datasets? Clarity would be substantially increased by a comparison chart that enumerates the attributes, advantages, and disadvantages of different instruments.

The detection of fusion-derived circRNAs is the focus of INTEGRATE-Circ and INTEGRATE-Vis. The text doesn't, however, explain how their visualisation techniques vary or whether using them both at once is required. Readers who are unfamiliar with these tools may become confused by this.

Although SeqCVIBE was previously discussed in relation to circular RNA-seq, it is explained in full in this section. Redundancy results from repeating this tool's features without highlighting how specifically it relates to lncRNA analysis. It's unclear if lncRNA analysis was the original purpose of SeqCVIBE or if this is a supplementary use case.

The section explains PANDORA as a differential gene expression analysis p-value combination algorithm, however it doesn't go into enough detail about how it stacks up against other statistical techniques. Does it work better in lncRNA-specific settings than other approaches? Which datasets have confirmed its usefulness?

Given that FastQC and MultiQC are general-purpose programs that may be used in a variety of RNA-seq investigations, their mention for quality control is a little out of place. Unless specifically linked to lncRNA-specific difficulties, their inclusion in a lncRNA-focused section does not significantly offer value.

The section highlights JBrowse integration for lncRNA visualization, however it doesn't explain why this visualization technique is so well-suited for lncRNA study. A more tangible illustration of how JBrowse facilitates new discoveries of lncRNAs would enhance the conversation. The section might have included well-known visualization issues specific to lncRNA research, such how to deal with sparse expression data or how to identify regulatory components in lncRNA sequences.

There is no obvious link between the results of tools like Integrative Genomics Viewer (IGV) and biological conclusions. How do theories regarding the regulatory roles of lncRNA or its connection to illnesses like cancer get informed by IGV's visualizations?

Phrases like "careful planning and validation" and "extensive datasets, creating logistical and computational bottlenecks" are ambiguous. Clarity could be gained by giving specific examples of these difficulties, such as problems with reproducibility, processing costs, or data storage.

There are no specific instances of current techniques or prototypes that support the recommendation to incorporate artificial intelligence (AI). Though unexplored, the mention of AI-enhanced workflows (such spatial transcriptomics) is promising.

Although the usefulness of Kaplan-Meier survival plots in cancer is mentioned, they are introduced abruptly and without any prior explanation of how they relate to RNA-seq data. Coherence would be enhanced by a link between survival analysis and RNA-seq results.

Although the article stresses the need of user-friendly interfaces, it makes no assessment of whether the tools that are highlighted such as scViewer and dittoSeq meet these requirements. The argument would be strengthened by usability information or instances of effective adoption in clinical labs.

Minor revision

Studies spanning 2017 to 2024 are included, according to the review. Unless they are early-access publications, it might be dubious to include studies from 2024 given the submission deadline of November 2024. Explain how 2024 research was included.

Uncertainty surrounds the quoted percentages for various RNA-seq data types (19% for circular RNA-seq, 56% for single-cell RNA-seq, etc.). Does this ratio apply to all identified studies (126) or to the total number of included studies (33)? Give specifics to ensure clarity. Practical use appears to be at odds with the percentage breakdown of RNA-seq kinds (e.g., 18% circular RNA-seq). For instance, the use of circular RNA-seq is not as widespread as suggested. Support these numbers with evidences.

Repetition occurs because the RNA-seq benefits section restates information from the abstract and subsequent parts. For example, bulk RNA-seq and single-cell RNA-seq are described more than once.

"in-vivo" and "in vitro" ought to be capitalized consistently, ideally italicized as "in vivo" and "in vitro."

A misuse of the term "in silico" has occurred. Another way to say it would be "computational analysis of RNA-seq data."

Instead of 39 as line 233 states, line 230 lists 22 studies for single-cell RNA-seq, 5 for bulk RNA-seq, 7 for circular RNA-seq, and 1 for long non-coding RNA. Explain this seeming contradiction.

The phrase “that’s why” in line 233 is not formal. For a more official tone, think about "which is why" or "resulting in the total".

Grammatical errors can be seen in line 284: "Li et al. [41] developed scRNASequest, provides an end-to-end solution...". Rewrite such that "Li et al. [41] developed scRNASequest, which provides an end-to-end solution…" .

The sentence "Though Searchlight is subjected as the good platform" (line 314) is confusing and grammatically wrong. Replace "Though Searchlight is regarded as a good platform."

Both the Bulk RNA-seq (lines 331–336) and Single-cell RNA-seq (lines 260–264) sections feature tools such as dittoSeq. While cross-referencing these overlaps could eliminate redundancy and improve clarity, this makes sense for tools that handle different data types.

A little ambiguous is line 277: "Mapper’s graph-based representations preserve the continuity in the data...". Describe in further detail how Mapper accomplishes this, or give a clear example.

Think about adding more case studies or real-world application examples for programs like Mapper and DTAE.

Line 308: While certain approaches, like PCA plots, are given in brevity, others, like MA plots and volcano plots, are explored in greater length. Clarity would be enhanced by balancing these descriptions.

Line 350: The Force-Directed Multilevel Maximally Modular (FMMM) algorithm is described in technical terms, but its biological applicability is not adequately explained. Simplify or briefly explain the scenario.

Line 355: "Static and dynamic cellular pictographs" are mentioned in the text, but the distinction between them is not explained. Adding visuals or instances would improve comprehension.

Line 360: It's unclear how important GoM models are in comparison to more conventional clustering techniques. It would be beneficial to include a quick example or comparison.

Redundancy results from the mention of visualization techniques like PCA plots, UMAP, and heatmaps in many tools. Simplify these descriptions to highlight unique characteristics.

While some programs, like Graphia Professional, are defined more technically, others, like BEAVR and SeqCVIBE, are described in terms of their user-friendliness. A uniform methodology for tool evaluation would improve comparability.

The abbreviation is presented without explanation, despite the fact that DEGs are essential to RNA-seq investigations.

Despite their clinical implications, tools such as "Sankey diagrams" and "pathway enrichment plots" are not adequately explained in terms of how they are created or evaluated in relation to RNA-Seq data.

Style inconsistencies are caused by the hyphenation and inconsistent capitalization of "RNA-seq" and "RNA-Seq" throughout the text.

The integration of RNA-Seq data with EHRs is mentioned in passing, but it is not further discussed, which means that there is a chance to investigate how this connection might enhance clinical workflows.

Although it is acknowledged that the inclusion of solely English-language studies has limitations, neither the geographic locations nor the languages that were left out nor how this limitation can affect the results are addressed.

Author Response

We thank the reviewer for sharing the insightful suggestions for our manuscript and providing feedback of our manuscript. Based on the reviewer’s comments received, we have proposed to make significant changes to the manuscript to address the weakness identified as described below with a point-by-point response to the reviewers’ comments and a description of changes proposed to be made to the manuscript.  All the changes are incorporated in the paper with highlighted text in yellow and the detailed responses in the attached pdf file.

Reviewer 3 Report

Comments and Suggestions for Authors

First of all there are hundreds of reviews about general RNA-seq and specific RNAseq applications. To find a new niche in this field requires to have a great experience in analysis of RNA data. Here authors describe some tools for working with RNA-sec and describe some basic principles, which exist already in many papers.  Authors selection is based on a simple PubMed search, but must be on the authors’ experience. This way authors are missing a lot of useful papers. Galaxy as the best example. 

Sections 1.1, 1,2  of the introduction are described hundreds of times elsewhere and much better. Difference between the first two Table 1b is not clear (Total RNA Sequencing, mRNA Sequencing). These are actually the same. 

“The search terms included various combinations of keywords related to "RNA-seq" and "visualization," connected using Boolean operators "OR"  “  - Are authors indeed believe that the readers are interested in how to make PubMed searches with boolean and /or ? 

"and alignment to reference genomes". - not true, sometimes there is no reference genome.

Users can modify visual elements like font sizes, axis labels, and colors - Are you really believe these are great features of programs ?

I tried SeqCVIBE, but could not make it work. That means authors did not try the described methods themselves. Additionally, this program developed exclusively to manage internal data in the Greek institutional network, works with preloaded datasets and human and mouse genomes only and has no means of uploading user data. Unfortunately authors even did not check the applicability of the tools they are describing.

Lastly, visualization is not so important point to write a review, it can be done in many ways, once the data is analyzed.

Author Response

General Comments: First of all there are hundreds of reviews about general RNA-seq and specific RNAseq applications. To find a new niche in this field requires to have a great experience in analysis of RNA data. Here authors describe some tools for working with RNA-sec and describe some basic principles, which exist already in many papers.  Authors selection is based on a simple PubMed search, but must be on the authors’ experience. This way authors are missing a lot of useful papers. Galaxy as the best example. 

Response: We thank the reviewer for sharing the insightful suggestions for our manuscript and providing feedback of our manuscript. Based on the reviewer’s comments received, we have proposed to make significant changes to the manuscript to address the weakness identified as described below with a point-by-point response to the reviewers’ comments and a description of changes proposed to be made to the manuscript. All the changes are highlighted in the revised version of manuscript. The detailed responses are in the attached file. 

Round 2

Reviewer 2 Report

Comments and Suggestions for Authors

The article has been updated in accordance with the modifications suggested, and the detailed answers are compelling. I don't think any more changes are required. Good luck.

Author Response

We thank the reviewer for reviewing our revisions and sharing a good feedback after our revisions. Thanks for your time and providing great comments top improve the manuscript.

Reviewer 3 Report

Comments and Suggestions for Authors

As I mentioned before, reviews must be based on the author's experience in the topic, but not merely on google-like search.  What is presented is a retelling, but not a critical review.  Lots of false statements in the work shows that the authors have no experience in RNA-seq analysis, nor in biology.

In the first review I pointed out the total irrelevance of the text and gave some examples of false statements. Authors tried to reply on this,  but the basics remained the same - authors have no experience in RNA sec. Which resulted in more new severe falls statements and old ones which I have not mentioned.  This cannot be accepted. 

Again only some issues: 

Right in the beginning: 

Every cell in our body contains the same set of genes - falls - only autosomal cells, and even that not in all organisms! 

biological activities (in-vivo), laboratory experiments (in vitro), - both can be in-vivo and in-vitro, What does see no difference in these words. 

Sections, 1.1 -1.2; Table 1;  - Already described elsewhere many times 

MegaBLAST - Because authors have no experience they are not aware that BLAST bunch of programs is used for sequence alignment and not for visualization of RNA-seq. 

Fig S1 and S2 are essentially the same and I see no conceptional difference, while authors probably classified programs according to linguistic similarity disregarding the functionality. 

“A key limitation of this review is that we relied on tool descriptions provided in the literature or documentation and did not independently test all the tools due to time and resource constraints. ” - I have never experienced such before.  I would reword it as “we have no idea if the described tools actually exist or work, we solely retell the abstracts”. 

Author Response

General Comments: As I mentioned before, reviews must be based on the author's experience in the topic, but not merely on google-like search.  What is presented is a retelling, but not a critical review.  Lots of false statements in the work shows that the authors have no experience in RNA-seq analysis, nor in biology. In the first review I pointed out the total irrelevance of the text and gave some examples of false statements. Authors tried to reply on this,  but the basics remained the same - authors have no experience in RNA sec. Which resulted in more new severe falls statements and old ones which I have not mentioned.  This cannot be accepted. 

Response: We appreciate the reviewer for taking time to review our revised manuscript. Please find below our responses for some of the issues raised for our manuscript in the prior and current revision:

  • The manuscript is written on a mere google/ PubMed search

Author response: This manuscript is not written based on a google/ PubMed search. This paper is the collective 8- 9 months of authors’ efforts starting from March 2024 - Nov 2024. We started with identifying the research topic, designing research questions, finding the databases,  designing search query based on research questions, refining the search query, finding the articles,  developing a database of it and then following the process as mentioned in Figure 1. All of these steps can’t be accomplished with a mere google/pubmed search in a single day/month. Section 2 provides the complete details of our step by step process followed in this systematic literature review. All the details are complete, and thorough and provide evidence on our review process. 

  • The manuscript is not written based on the author's experience.

Author response: We appreciate reviewer’s knowledge and expertise in RNA-seq field which helped us in improving the  manuscript. However, we are not new in this field. Lead authors Farhana and Cyruss both have deep knowledge, expertise, and experience in biology and data science. Senior author Vibhuti is working in the field of biomedical science from last 10 years. 

  • The manuscript only retells the abstract but not a critical review. 

Author response: The abstract of a paper doesn’t contain the details of the tools functioning, its applications, and clinical use case. This requires reading and understanding of the full paper. This paper is the result of synthesis of 33 primary articles considered for full review. We have thoroughly read all the articles, summarized them, identified key pointers required to answer the proposed research questions, designed tables, critically analyzed the results to understand limitations and evaluate results. Based on our review and research experience, a critical analysis can’t be designed by reading the abstracts of the papers. 

  • Authors describe some tools for working with RNA-sec and describe some basic principles, which exist already in many papers.

Author response: We have thoroughly examined the review papers in the timeline provided in Figure 1 (March 2017 - Jan 2024) and found that few of the prior articles have the detailed review of the tools and techniques for RNA-seq data. To the best of our knowledge, our paper is unique in terms of below criteria:

                     Some of the reviews published in the scope of our review timeline  are as follows:

1) Clark, A. J., & Lillard Jr, J. W. (2024). A Comprehensive Review of Bioinformatics Tools for Genomic Biomarker Discovery Driving Precision Oncology. Genes, 15(8), 1036.

2) Rogé, X., & Zhang, X. (2014). RNAseqViewer: visualization tool for RNA-Seq data. Bioinformatics, 30(6), 891-892.

3) Bagnacani, A. (2023). Visualization of RNA-Seq results with CummeRbund. Galaxy Training Network.

4) Conesa, A., Madrigal, P., Tarazona, S., Gomez-Cabrero, D., Cervera, A., McPherson, A., Szcześniak, M. W., Gaffney, D. J., Elo, L. L., Zhang, X., & Mortazavi, A. (2020). Systematic comparison and assessment of RNA-seq procedures for gene expression quantification. Scientific Reports, 10(1), 19737

5) Conesa, A., & Mortazavi, A. (2024). RNA‐Seq Data Analysis: A Practical Guide for Model and Non‐Model Organisms. Current Protocols, 4(5), e1054

6) Batut, B., van den Beek, M., Doyle, M. A., & Soranzo, N. (2021). RNA-seq data analysis in galaxy. RNA Bioinformatics, 367-392.

  • Prior reviews(1,4,5) were focused either on the RNA-seq tools for specific disease conditions such as cancer or tutorial to demonstrate the full analysis of RNA-seq data. 
  • Articles 2 is not within the timeframe of our review and 3 is a tutorial. 
  • Article 6 is within our timeframe and a tool to analyze RNA-seq data however this paper is also a tutorial of using this tool. 
  • In the first review I pointed out the total irrelevance of the text and gave some examples of false statements. Authors tried to reply on this,  but the basics remained the same - authors have no experience in RNA sec. Which resulted in more new severe falls statements and old ones which I have not mentioned.  This cannot be accepted. 

Author response: We have tried our best to improve the manuscript following reviewers recommendations in the first revision as well as in the second revision. We appreciate the reviewer if more specific on the false statements in the text can be provided to us to improve the manuscript. 

Again only some issues: 

Right in the beginning: 

R3.1: Every cell in our body contains the same set of genes - falls - only autosomal cells, and even that not in all organisms! 

Response: We thank the reviewer for sharing this major comment with us. We agree with the reviewer that every cell in our body doesn’t contain the same set of genes and is with autosomal cells. To clarify and avoid confusion we used somatic cells to refer to all body cells and added exceptions to certain types of cells. We rephrased the sentence as shown below in Lines 20-22 and highlighted:

Every somatic cell in our body contains the same set of genes encoded in our genome, with the exception of gametes and certain specialized cells (i.e., red blood cells, gametes, and some immune cells etc.)

R3.2: biological activities (in-vivo), laboratory experiments (in vitro), - both can be in-vivo and in-vitro, What does see no difference in these words. 

Response: The biological activities (in-vivo) and laboratory experiments (in-vitro) both can be in-vivo and in-vitro depending upon the study settings. However, for RNA-sequencing both in-vivo and in-vitro aspects represent different stages of RNA-seq. Thus, both together is not possible for RNA-seq, however possible in other study settings. RNA-seq is a combination of in-vivo and in-vitro processes necessary to study the transcriptome. The manuscript is revised as shown in the highlighted texts in the current version.

R3.3: Sections, 1.1 -1.2; Table 1;  - Already described elsewhere many times 

Response: We significantly reduced the redundant part and made significant changes in the sections 1.1 and 1.2 of the paper as per the reviewer’s suggestion in our prior revision as well as in the current version. The highlighted part shows the revised sections 1.1 and 1.2 in the revised version. However if the the reviewer finds it again in our revised manuscript, specifics on the part of the manuscript are required for us to address it better. 

R3.3: MegaBLAST - Because authors have no experience they are not aware that BLAST bunch of programs is used for sequence alignment and not for visualization of RNA-seq. 

Response: We appreciate reviewer’s insights on sharing this comment with us and expertise in the field of RNA-seq. We agree that MegaBLAST is not a visualization tool and aware of it, however it was included by mistake in the table. As per reviewer’s suggestion, we removed MegaBLAST from Table 3. 

R3.4: Fig S1 and S2 are essentially the same and I see no conceptional difference, while authors probably classified programs according to linguistic similarity disregarding the functionality. 

Response: Figures S1 and S2 differ slightly in terms of their usage. Figure S1 represents the usage of the tools whether it is used in the browser, dynamically or using specific packages. However, S2 shows the distribution of tools by their development language and framework. Overall, in summary Figure S1 shows how the users can use these tools to analyze their RNA-Seq data however, Figure S2 shows which source language/platform is used to develop these tools.   

R3.5: “A key limitation of this review is that we relied on tool descriptions provided in the literature or documentation and did not independently test all the tools due to time and resource constraints. ” - I have never experienced such before.  I would reword it as “we have no idea if the described tools actually exist or work, we solely retell the abstracts”. 

Response: We truly appreciate the reviewer's knowledge and expertise in the field however the suggested rewording is not feasible for any research article as it undermines the authors' contributions and efforts in designing this work. To the best of our knowledge, the goal of a systematic review is to collect, evaluate, and synthesize existing research and provide reliable evidence for future research, which we are following and providing sufficient evidence of it. 

The purpose of this systematic review article is to inform a broad set of users including basic scientists, lab staff, bioinformaticians, and computer scientists aware of various visualization tools and techniques used to analyze and interpret RNA-Seq datasets so that they can choose the tools for their specific clinical use case and know about the advantages and limitations of each of them. We are only focusing on addressing below research questions in this article as mentioned in Section 1.3 (Lines 134-139) and shown below:

  • What are the primary visualization tools and techniques currently employed in RNA seq data analysis, with a particular focus on clinical interpretations? 
  • How do these visualization tools enhance the understanding and interpretation of  RNA-seq data within the clinical realm?
  • What improvements can be made to existing tools, or what new tools are required, to enhance the utility and effectiveness of RNA-seq analyses in clinical applications?

Any question outside the scope of above research questions will not align with the goals of this paper. The testing of each of the tools is outside the scope of our paper and is a limitation of this work too as we mentioned it in the paper. We will definitely try to accomplish testing of tools in our future work but the article will have different goals to empirically provide evidence on the usage of these tools not to provide an overview of existing tools available for visualizing and interpreting RNA-Seq data. 

Round 3

Reviewer 3 Report

Comments and Suggestions for Authors

In the first review I pointed out the total irrelevance of the text and gave some examples of false statements (out of very many). Authors tried to reply on this,  which resulted in more new false statements, which I pointed out in the 2nd round. But the basics remained the same - authors have no experience in RNA sec and this is not a critical review. Inclusion of nonrelevant and not working programs proves that.

For the clinical community, especially physicians, scientists, and laboratorians who might not be deeply versed in the nuances of computational biology, - you are contradicting yourself - physicians with no computational skills you explain tools in R scripts and Command line tools under Linux. Are you sure physicians can do R scripting?

As before, some false statements out of many:

RNA-seq has transformed the field of genomics since its inception by enabling the sequencing of entire transcriptomes at a fraction of the time and cost of earlier methods. - Arrays were cheaper and faster! 

RNA-seq can provide a wide dynamic range by detecting…  107

Additionally, RNA-seq has a wide dynamic range, detecting… - repeated for better explanation? 

Some comments on your prev reply:

If you have an “expertise, and experience in biology and data science” write a review in data science. In this topic your expertise is far from being sufficient. 

describe some basic principles, which exist already in many papers.  - this relates to your sections  1.1. 1.2 - described better before many times. Start from 1.3.

designing search query - are you kidding me? Several months to “combine keywords related to "RNA-seq" and "visualization," connected using Boolean operators "OR" and "AND" “ ? 

 finding the articles - is that no more than two mouse clicks ? 

“ All of these steps can’t be accomplished with a mere google/pubmed search in a single day/month.” - One week maximum. In any case the amount of time you spent on designing a search query has no value on the article's positive evaluation .

“To the best of our knowledge, the goal of a systematic review is to collect, evaluate, and synthesize existing research “ - that is absolutely right, but you have none of these, to be able to do that, you need knowledge in the topic and practical experience. Inclusion of MegaBlast and non-running programs is MISLEADING the readers (especially physicians)